

# Encoder-decoder convolutional neural network for simple CT segmentation of COVID-19 infected lungs

Kiri S. Newson[1], David M. Benoit[2] and Andrew W. Beavis[3,4,5]

[1] Department of Physics and Mathematics, University of Hull, Hull, United Kingdom
[2] E. A. Milne Centre for Astrophysics, Department of Physics and Mathematics, University of Hull, Hull, United Kingdom
[3] Medical Physics Department, Queen's Centre for Oncology, Hull University Teaching Hospitals NHS Trust, Cottingham, Hull, United Kingdom
[4] Medical Physics and Biomedical Engineering, University College London, University of London, London, United Kingdom
[5] Hull York Medical School, University of Hull, Hull, United Kingdom

Corresponding author
Kiri S. Newson,
k.s.newson-2016@hull.ac.uk

## ABSTRACT

This work presents the application of an Encoder-Decoder convolutional neural network (ED-CNN) model to automatically segment COVID-19 computerised tomography (CT) data. By doing so we are producing an alternative model to current literature, which is easy to follow and reproduce, making it more accessible for real-world applications as little training would be required to use this. Our simple approach achieves results comparable to those of previously published studies, which use more complex deep-learning networks. We demonstrate a high-quality automated segmentation prediction of thoracic CT scans that correctly delineates the infected regions of the lungs. This segmentation automation can be used as a tool to speed up the contouring process, either to check manual contouring in place of a peer checking, when not possible or to give a rapid indication of infection to be referred for further treatment, thus saving time and resources. In contrast, manual contouring is a time-consuming process in which a professional would contour each patient one by one to be later checked by another professional. The proposed model uses approximately 49 k parameters while others average over 1,000 times more parameters. As our approach relies on a very compact model, shorter training times are observed, which make it possible to easily retrain the model using other data and potentially afford "personalised medicine" workflows. The model achieves similarity scores of *Specificity (Sp)* = 0.996 ± 0.001, *Accuracy (Acc)* = 0.994 ± 0.002 and *Mean absolute error (MAE)* = 0.0075 ± 0.0005.

# INTRODUCTION

The severe acute respiratory syndrome 2 (SARS-CoV-2) virus causing the disease COVID-19 (*Ciotti et al., 2020*), was first reported in January 2020 in the UK (*Flynn et al., 2020*). This disease was spread *via* "person-to-person transmission" (*Flynn et al., 2020*) and soon declared a pandemic by the World Health Organisation (WHO) on 11th March 2020 (*Ahishali et al., 2021*). Since no effective vaccines were available early in the pandemic and

without an actual cure for COVID-19, it quickly became vital to diagnose cases to try and stop the spread of the disease. One way to do this was to detect early signs of infection and isolate that individual from the rest of the population (*Ai et al., 2020*).

While current diagnosis methods of COVID-19 rely on reverse transcription-PCR (RT-PCR), a chest computerised tomography (CT) scan was used early in the pandemic to diagnose infection prior to a positive RT-PCR test (*Adil et al., 2021*). Indeed, *Ai et al. (2020)* states that chest CTs may be used as a primary tool for COVID-19 detection. However, COVID-19 infections can appear similar to other viral pneumonia on chest CTs (*Ai et al., 2020*), thus leading to false positives using this method. In the present work (see Methods) we assume that all detected infections in both training and testing datasets are confirmed cases of COVID-19.

We show typical examples of infection areas on lung CT in the Materials subsection in the Methods section. Note that specialist knowledge is required to generate the initial contours that delimit the affected areas, a process that can be time-consuming and requires extensive high-level training. When looking at similar scenarios where contours are required, such as diagnosing or treating those with lung cancer, it is common practice, following the guidelines given by *The Royal College of Radiologists (2022)*, to have these contours peer-reviewed by professionals. This can add significant delays to the process. Nonetheless *Vaz et al. (2022)* explains how the additional peer review is important, as it is one of the main uncertainties when treatment planning patients with lung cancer. Given the large number of patient admissions in times of pandemic, an automated method to identify infected areas would help free up time for healthcare specialists. Note that we are not advocating the replacement of experts but instead a faster workflow where the systems contours to be reviewed. This would allow more patients to be examined/diagnosed and ultimately improve treatment outcomes/survival rates overall for affected patients. The method introduced in this article can easily be adapted to other lesions/pathologies detected using CT imaging (*e.g.*, pneumonia, cancer, emphysema, embolisms) which could be used as a tool in the future to second check contours, especially in situations where a peer is not readily available to check the contours, again speeding up the process. This would be particularly valuable in underfunded hospitals with limited time and equipment to give patients the best chance of survival with a faster diagnosis or treatment plan.

Several studies have explored the use of machine learning (in particular deep learning) to what is technically a segmentation problem in the medical field. Many of these used adaptations of the U-Net structure by *Ronneberger, Fischer & Brox (2015)*, which links in with the chosen architecture for this work (see Methods). One of these studies which used a U-Net model is that of *Raj et al. (2021)*, their study showed that the standard method used for automatic detection on CT scans has issues with high variations in intensities and indistinct edges near the infected regions. Moreover, they also discussed the influence of noise on actual detection, likely originating from the data acquisition process. *Raj et al. (2021)* proposed a new COVID-19 pulmonary infection segmentation depth network which they referred to as attention-gate dense network improved dilation convolutional U-Net (ADID U-Net). Their results show that their model can accurately segment COVID-19-infected areas in lung CTs with a performance of over 80% for their chosen

metrics (accuracy measure, specificity measure, and dice coefficient, see evaluation metrics subsection for details. Similarly, *Elharrouss, Subramanian & Al-Maadeed (2021)* show successful segmentation of COVID-19 in CT scans using their two-stage model of an encoder-decoder along with the added step of pre-processing to extract structure and texture of the images. The pre-processed images are then used as the inputs of the encoder-decoder which is based on the SegNet model (*Elharrouss, Subramanian & Al-Maadeed, 2021*).

More studies follow the basic structure of the U-Net model with adaptations to segment other medical images such as with breast ultrasound (BUS) images, which are the images produced when screening patients for breast cancer. *Zhuang et al. (2019)* proposed a Residual-Dilated-Attention-Gate-U-Net (RDAU-NET) model to segment tumours in (BUS) images. Although this is based on the U-Net model, it replaces plain neural units with residual units which allows for enhanced edge information which is an issue for many medical images, especially ultrasound images. Others who look at models specifically for segmentation of BUS images include the use of RDA-UNET-WGAN with the use of Wasserstein generative adversarial networks (WGANs) (*Negi et al., 2020*). This approach used generative adversarial networks (GANs) made up of two networks, a generator and a discriminator network. Their RDA-UNET model is used as the segmentation model (generator) and a fully connected convolutional neural network (CNN) is used as the discriminator to estimate the authenticity of the sample given which is what gave them the (RDA-UNET-WGAN model). Another U-Net based model is that of *Zhang et al. (2023)* which used a network made of two branches to segment BUS images. This comprises a classification branch and a segmentation branch. Both of which share the encoder layer and segmentation part of their network is based on the typical U-Net structure. Others, such as *Fan et al. (2020)*, have used an Inf-Net approach to segment COVID-19 lung infections which consists of "three reverse attention modules connected to paralleled partial decoder (PPD)".

In the present study, we investigate the hypothesis that a simpler deep-learning model can provide a suitable alternative to U-Nets which are a common framework for segmentation models. We suggest the usage of a convolutional encoder-decoder network which is structurally similar to an auto-encoder. The model's objective is to analyse unseen chest CT scans and produce a contour that delineates areas of COVID-19 infection within the lung. The training of our model requires a supervised scheme, contrary to the usual unsupervised auto-encoder neural network (AE-NN) training protocols since we are effectively transforming the initial CT scan into a segmentation mask. We then use the same metrics as the literature mentioned earlier to evaluate our model's performance. By doing so we are producing an alternative model to current literature which is easy to follow and reproduce making it more accessible for real-world applications as little training would be required to use this. Along with the model being accessible for others to train on their data, it is also a much simpler model with fewer parameters and has a relatively short training time (see Methods section). Should someone wish to train this model on their data, similar optimisation methods would be necessary to make it the best fit for a different dataset. Details of our approach are described in the Methods section, along with the

details of the dataset used. We present the results of our approach in the Results section before evaluating its performance in the Discussion section.

## METHODS

### Materials

We use Python (v3.7.15) and Keras (v2.9.0) to implement our model and use publicly available COVID-19 infection data[1] (data from *Jun et al., 2020*). This dataset, collected in April 2020, consists of 20 labelled full chest CT scans of COVID-19-infected lungs and their corresponding segmentation masks. These are labelled by two radiologists and verified by an experienced radiologist (*Jun et al., 2020*).

The white regions in Figs. 1B and 1D are the contours/masks drawn by healthcare specialists to indicate the areas affected by COVID-19. The corresponding original CT scan slices are shown on the left (Figs. 1A and 1C).

### Model

Our model is based on a convolutional neural network (CNN), a specialist type of neural network designed to process 2D image data. In this type of network, the CNN layers effectively convert the image into feature activation maps using tuneable filters (convolutional kernels) and thus can identify specific spatial patterns within a given image. The overall structure of our model is shown diagrammatically in Fig. 2. Its main architecture follows the shape of an auto-encoder (AE), which was chosen due to its similarities to the already successful U-Net structures. Both types of models have encoder and decoder sections, but the U-Net has additional connections between the two sections (*Karimov et al., 2019*). We suggest here that auto-encoders are theoretically able to carry out similar tasks to the U-Nets but provide a simpler and faster route to solutions without the complexity of a U-Net model.

Auto-encoders have seen a rise in popularity and have been prevalent in recent literature; such as those reported by *Shvetsova et al. (2021)* to detect abnormal sections of imaging in chest x-ray images, *Gong et al. (2019)* to carry out anomaly detection, *Abraham & Nair (2018)*, who used stacked sparse auto-encoders (SSAE) to classify prostate cancer into grade groups from MRI images and the work by *Qadri et al. (2023)* who also used SSAEs for automated vertebrae segmentation. Throughout the literature mentioned, each study uses a model based on an auto-encoder which prompted us to then develop a model which comprised a CNN with an encoder-decoder architecture to follow that of an auto-encoder.

The following explanations of the model and process to train the model, along with data augmentation are depicted in the flowchart in Fig. 3. This figure gives an outline of the whole process from reading the data to producing the prediction contour and will be referenced throughout the next section.

Following a supervised training protocol, the recorded lung CT scan is encoded through three padded convolutional layers into a compressed (latent) representation of the original CT image. We train the model to reconstruct not the original image, as would normally be the case for a standard AE-NN, but instead, the COVID-19-infected contour region using

---

[1] COVID-19 infected lung CT's and Masks are available at https://doi.org/10.5281/zenodo.3757476 which is accessed *via* https://www.medseg.ai/covid-19

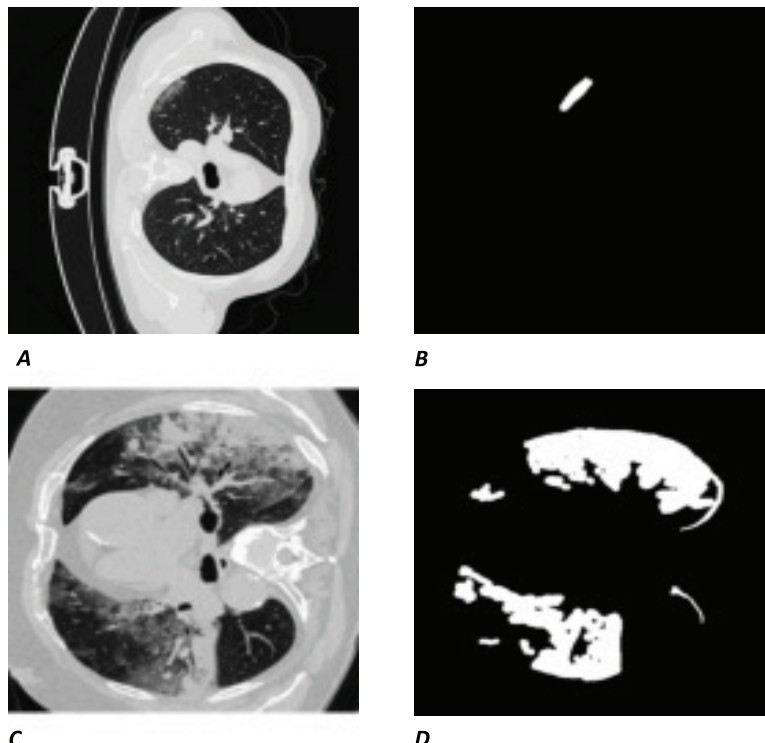

**Figure 1 Visual example of data used.** (A and C) Slices of a CT chest scan showing the cross-section of a lung infected with COVID-19; the grey areas showing where the infection is present. (B and D) The corresponding contours/masks, respectively, which are the segmented infected regions highlighting areas of the lung that are visibly infected. These masks are the "ground truth" (GT) used for training models and comparing predictions to. CT images/contours used within figure by *Jun et al. (2020)*.

its decoder section. The resulting image is a contoured version of the input CT that highlights infection areas.

The encoding section of our model (left of Fig. 2) and the "Encoder" section of Fig. 3, consists of three padded convolutional layers all with a ReLU activation function and $3 \times 3$ filters (padding = 'same'). The image size remains constant throughout the model (here $128 \times 128$ pixels) but the depth (number of filters) is halved as we progress through the encoder (from 64 to 32 then 16). The shape shown in Fig. 2 was investigated by looking at the loss curve for a range of filter sizes. Figure S1 in the supplementary information shows the result of testing different shapes of $(128 \times 64 \times 16)$, $(128 \times 32 \times 16)$ and finally $(64 \times 32 \times 16)$. We chose the $(64 \times 32 \times 16)$ filter version because the loss curve gives us minimal over-fitting compared to the other filter size combination.

The convolutional layers are then followed by a $(2 \times 2)$ max-pooling layer that effectively constitutes our compressed, latent, representation which can be seen between the encoder and decoder in Fig. 3 and at the centre of the diagram in Fig. 2. This max-pooling layer aims to extract the largest value for each $(2 \times 2)$ section of the feature maps and reduce the size of the input picture. Our latent $(64 \times 64 \times 16)$ representation is then up-sampled through 3 ReLU-activated de-convolutional layers in reverse order $(16 \times 32 \times 64)$ to reconstruct the COVID-19-infected contour region which can be seen

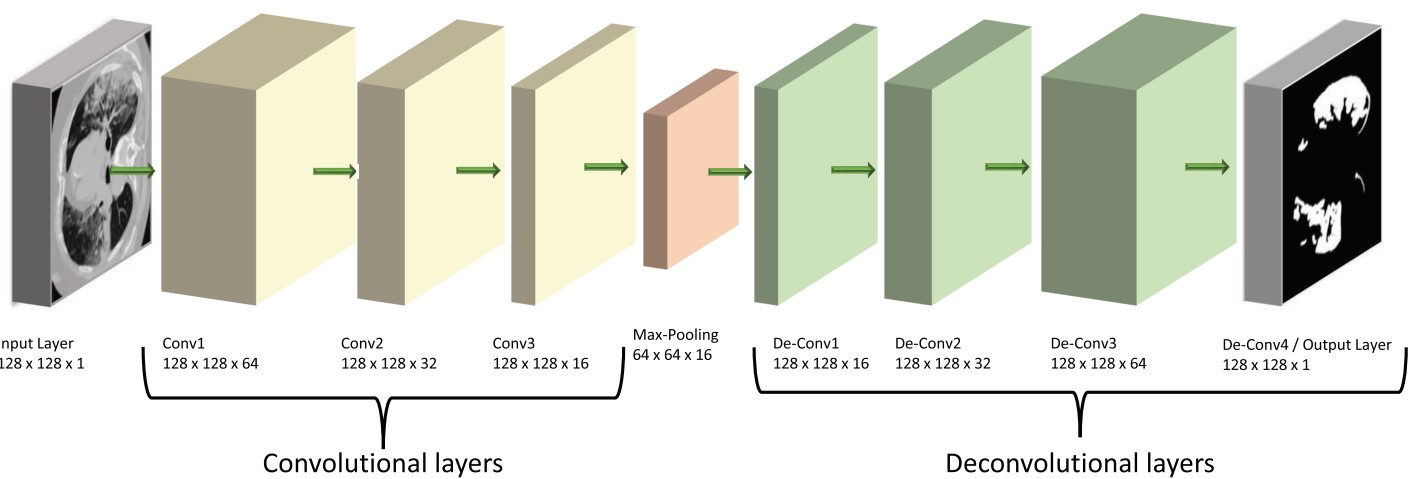

Input Layer
128 x 128 x 1

Conv1
128 x 128 x 64

Conv2
128 x 128 x 32

Conv3
128 x 128 x 16

Max-Pooling
64 x 64 x 16

De-Conv1
128 x 128 x 16

De-Conv2
128 x 128 x 32

De-Conv3
128 x 128 x 64

De-Conv4 / Output Layer
128 x 128 x 1

Convolutional layers

Deconvolutional layers

**Figure 2** **Diagram of our ED-CNN-** The recorded lung CT scan (left) is progressively encoded as it passes through the encoder. The compressed representation is then decoded into an output contour of the COVID-19 infected region(s) (right) of the input CT. CT images/contours used within figure by *Jun et al. (2020)*.

on the right side of Fig. 2 and in the "Decoder" section of Fig. 3. These de-convolutional layers are also padded (padding = 'same') with a stride set to two for the first layer, to recover the original size of the image. The final decoded layer uses a sigmoid activation function to generate the predicted contour region.

We optimise the binary cross-entropy loss function of the model using the Adam optimiser. In this case, the binary cross-entropy is particularly well suited as the ground truth (GT) contours are given as black and white (0/1) pixels. Effectively, this translates to a pixel-by-pixel truth table that indicates COVID-19 infection or lack of it throughout the image. Other loss functions, such as mean squared error (MSE) for example, do not provide easily optimisable landscapes as most of the image is black with small white regions.

The dataset used originally held 3,520 images (each slice from the full CT creates one "image") of both the CT and corresponding mask. All the blank masks were then removed so the model was purely trained on images with COVID-19 present which can be seen in the "Data Pre-processing" section of Fig. 3. After the blank masks and corresponding CTs were removed, we were left with 1,844 images that could then be randomly split into test and training sets. We chose to split this into 20% (369 images) as test data, which was set aside for later evaluation of the model. It is crucial to split the data before any augmentation or manipulation to make sure it is a true test of the model and ensure the model has not seen another version of the image in the training process. The remaining 80% of the data (1,475 images) was used for training. First, we wanted to increase the training dataset to match more closely to the larger amounts of data that others have used such as *Raj et al. (2021)*, who used a data size of 9,226 images and (*Elharrouss, Subramanian & Al-Maadeed, 2021*) who used a data size >2,000 images. Our training data was therefore augmented by performing mirror operations (left/right and up/down),

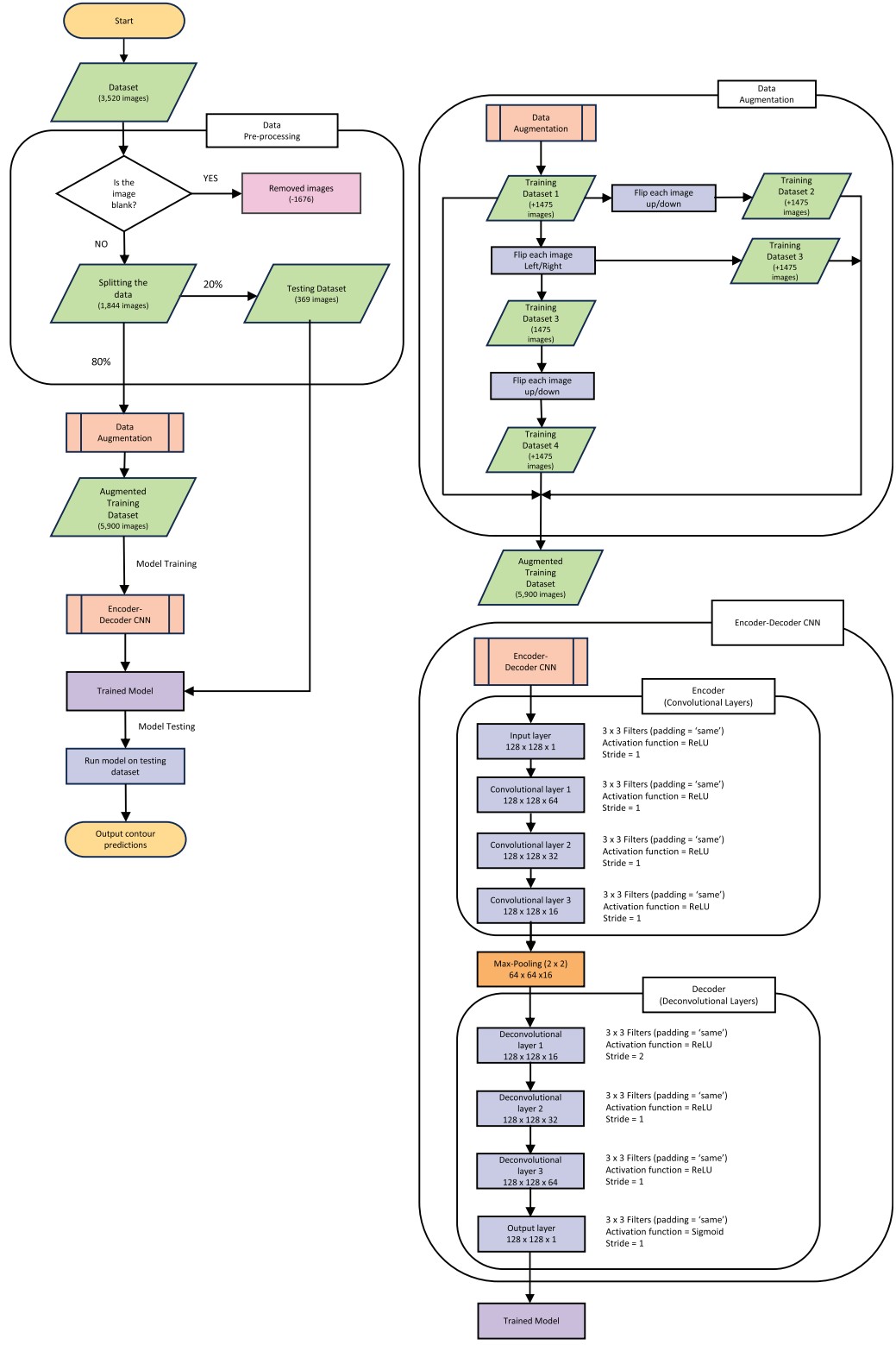

**Figure 3 Pipeline flow chart of the entire system.** Diagram includes the data augmentation process and the steps throughout the model.

following *Gandhi (2021)*. The process is depicted in Fig. 3 which shows the stages used to achieve the larger training dataset of 5,900 images. A visual representation of how the data was augmented can be seen in Fig. S2 in the supplementary information. Note that this is still a small dataset compared to some other training dataset sizes used in other published studies mentioned previously.

The new Augmented dataset of 5,900 images is then used to train the model which runs through the steps in the "Encoder-Decoder CNN" section of Fig. 3. To train our ED-CNN model, we use an input CT and an output of infected contours/segmentations of the corresponding CTs. This training had a batch size of two and trained for 60 epochs. A hyperparameter we investigated was the optimal number of epochs for our model, much like choosing the best filter previously when looking at the overall shape of the model. The model was trained using different numbers of epochs; 40, 50, 60, 120 and 240 epochs, Fig. S3 in the supplementary information shows this process and its results. We found 60 to be the best number of epochs before over-fitting starts to occur, preventing the model from learning unwanted noise (*Das & Das, 2023*). After the ED-CNN is trained, the prediction is run along with a visual representation of the CT and corresponding predicted contours for the "Testing Dataset" in Fig. 3, which was kept separate during the training process. This then gave us the output of a predicted contour to compare to the GT contours created by the healthcare professional to determine how good the prediction was.

## Evaluation metrics

We first use three popular metrics to compare our predicted segmentation to the ground truth. These metrics are described below and are also used in other studies such as *Elharrouss, Subramanian & Al-Maadeed, 2021*, *Fan et al. (2020)* and *Raj et al. (2021)*. These were run using the MATLAB code from *Raj et al. (2021)* which is publicly available and linked in their "data availability" section. Symbols used throughout are summarised in Table A1.

### *Structural similarity measure (structure-measure)*

*Fan et al. (2017)* put forward a new metric for evaluation which combines object-aware and region-aware measurements to form one measurement. As stated by the name, this measures the structural similarities between a ground truth mask and its prediction mask/segmentation and gives a result closer to how human vision works (*Fan et al., 2020*).

$$S_m = S_\alpha = (1 - \alpha) \cdot S_o(S_p, G) + \alpha \cdot S_r(S_p, G) \tag{1}$$

Here, $\alpha$ is a constant set to 0.5, $S_o$ represents the target perception similarity, $S_p$ the final prediction result, $G$ is the ground truth mask and $S_r$ the regional perceptual similarity (see also *Fan et al. (2017)*, *Cheng & Fan (2021)*, *Raj et al. (2021)*). As this measure is widely used by those mentioned along with *Fan et al. (2020)* it allows for a fair comparison to other studies.

### Enhanced—alignment measure (E-measure)

The enhanced alignment measure, also proposed by *Fan et al. (2018)*, combines the local pixel values with image-level mean values in one term This is given by Eq. (2) below:

$$E_a = Q_{F_M} = \frac{1}{w \cdot h} \sum_{x=1}^{w} \sum_{y=1}^{h} \phi(S_p(x,y), G(x,y)) \tag{2}$$

where $w$ and $h$ denote the width and height of $G$, $(x,y)$ are the coordinates of each pixel in $G$ and $\phi$ is the enhanced alignment matrix (see also *Fan et al. (2018)* and *Raj et al. (2021)*). The measure evaluates both local and global similarities when considering two binary maps (*Raj et al., 2021*).

### Mean absolute error (MAE)

This is a popular measure which evaluates pixel-wise error between $S_p$ and $G$ (*Fan et al., 2020*), given by Eq. (3).

$$MAE = \frac{1}{w \cdot h} \sum_{x}^{w} \sum_{y}^{h} |S_p(x,y) - G(x,y)| \tag{3}$$

The following metrics were also used, based on confusion matrices, when comparing the ground truth image and our prediction image. The following metrics make use of the four outputs of the confusion matrix: true negative (TN), true positive (TP), false negative (FN), and false positive (FP), each of which is explained below.

### Confusion matrix components

The four components that make up the outputs of the confusion matrix required for the further metrics used are explained by the following:

TP = Predict infected region and it is True (correct)
TN = Predict no infection region and it is True (correct)
FP = Predict infected region and it is False (incorrect)
FN = Predict no infection region and it is False (incorrect).

### Accuracy (Acc)

This is the ratio of correctly predicted pixels to the total number of pixels present given by Eq. (4).

$$Acc = \frac{TP + TN}{TP + TN + FP + FN} \tag{4}$$

### Precision (Pc)

This considers only the infected region pixels (the white areas) and gives us the ratio of correctly predicted infected pixels to the total number of predicted infected pixels. This is given by Eq. (5).

$$Pc = \frac{TP}{TP + FP} \tag{5}$$

### Sensitivity (Sen)

This gives us the ratio of correctly predicted infected pixels to the total number of actually infected pixels (the ground truth of infected pixels), given by Eq. (6).

$$Sen = \frac{TP}{TP + FN} \tag{6}$$

### F1 score (F1)

This combines both the Precision and Sensitivity in Eqs. (5) and (6) to give accuracy. This is given by Eq. (7).

$$F1 = 2 \frac{Pc \cdot Sen}{Pc + Sen} \tag{7}$$

### Specificity (Sp)

This gives us the ratio of correctly predicted non-infected region pixels (black areas) to the total number of actual non-infected pixel regions, given by Eq. (8).

$$Sp = \frac{TN}{TN + FP} \tag{8}$$

A mixture of these metrics was commonly used amongst the literature mentioned previously such as *Elharrouss, Subramanian & Al-Maadeed, 2021*, *Fan et al. (2020)*, *Raj et al. (2021)*, *Zhuang et al. (2019)* and *Negi et al. (2020)* which will be used to compare to our model in the results section.

## RESULTS

### Qualitative results

To test the performance of our ED-CNN, we used a randomly selected subset of our dataset (20%) prior to augmentation, which was kept aside as the testing dataset. The testing dataset consists of 369 images of both lung CT scans and corresponding infected masks/contours, all of which are unseen by the model. If the performance of the model is good, it will produce predicted images of the masks close to the corresponding GT images of the infected masks/contours of the CT scans, thus successfully indicating the areas of infection. Typical-generated contours, along with the original CT scans and the corresponding ground truths are shown in Fig. 4. Visually, we can see that our model can successfully predict the contours of the infected regions of the lungs with good accuracy as can be seen by comparing the "Pred" row, to the "GT" row in Fig. 4.

### Quantitative results

Whilst looking at the images in Fig. 4, it may be difficult to see the differences that occur between the predictions of our model and the GT images (second and third row respectively). To show more concretely that these images are similar, we rely on similarity scores and evaluation techniques that compare the similarities between the two images. The predicted images created are evaluated by testing with our 369 image testing dataset,

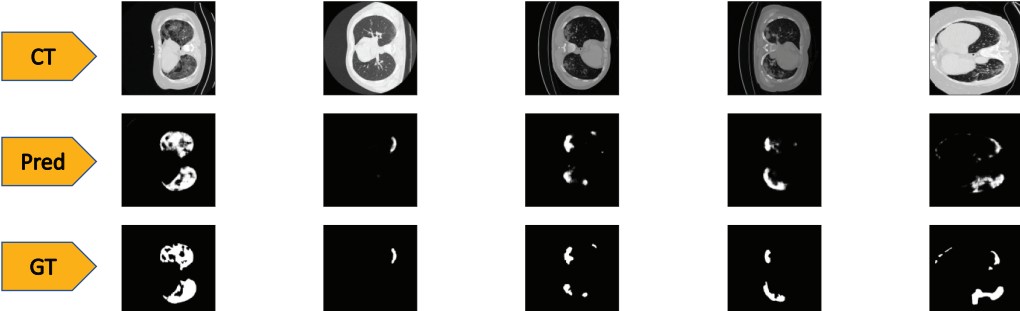

**Figure 4** **Visual comparison of results**- The first row shows the slices from a CT scan of the lungs that are infected with COVID-19, the second row shows the predicted infected regions made by our ED-CNN, the white highlighting the areas of infection from the original CT, and the final row shows the ground truth of the infected areas for comparison which also highlights infected areas in white-these are contoured by a radiologist. For rows one and three, CT images/contours used within the figure by *Jun et al. (2020)*.

against the corresponding GT images using the metrics described in the Methods section. For the structural similarity metric Eq. (1), a score close to 1 indicates a better likeness to the ground truth, with 1 being an exact match. Similarly with the metric, $E_a$ represented in the Eq. (2), the closer this result is to the value 1, the closer both images are said to be in similarity. The measurement for MAE given in Eq. (3) differs in that the value for the images to be close in similarity the score would tend closer to 0. The remaining metrics of Sen, Sp, Pc, F1, and Acc are all ratios, so that values closer to 1 indicate a higher similarity between the prediction and ground truth images.

The numerical results from our chosen metrics are given in Table 1 and we also summarise the type of model each of the comparative literature uses in the method column.

We compare our scores against six published models: (*Elharrouss, Subramanian & Al-Maadeed, 2021*; *Fan et al., 2020*; *Raj et al., 2021*; *Zhuang et al., 2019*; *Negi et al., 2020*). Table 1 shows that our ED-CNN score for $S_m$, is $0.82 \pm 0.01$. This is higher than the results in *Fan et al. (2020)*, but slightly lower than the two models from *Raj et al. (2021)* score. This highlights that our model produces predictions that compare well with more elaborate models such as the ADID-U-Net by *Raj et al. (2021)*, for example. When looking at the $E_a$ scores, our models score is $0.86 \pm 0.01$, which does fall below the other models but not by a large amount. For *MAE* our model achieved $0.0075 \pm 0.0005$, better than all the comparable literature (*Elharrouss, Subramanian & Al-Maadeed, 2021*; *Fan et al., 2020*; *Raj et al., 2021*; *Zhuang et al., 2019*; *Negi et al., 2020*), as it is closer to 0 compared to their scores. Our ED-CNN also scored best for $S_p$ and *Acc* when compared to the others in Table 1. For *Sen* our ED-CNN scored $0.82 \pm 0.05$ which is close to the highest score of 0.8837 by *Negi et al. (2020)* and higher than *Elharrouss, Subramanian & Al-Maadeed, 2021*, *Fan et al. (2020)*. Our *Sen* value is between both (*Raj et al., 2021*) and (*Zhuang et al., 2019*), within error bars. It is worth noting that our model achieves these scores with a low number of parameters (49 K for our ED-CNN), compared to the others ranging from 28,817 K (*Negi et al., 2020*) to 56,223 K (*Raj et al., 2021*).

**Table 1 Comparison of results.** The results compared to the performance of previous literature referenced in the table. We show the scores for eight commonly used metrics; ($S_m$) Structural similarity measure, ($E_a$) Enhanced-alignment measure, ($MAE$) mean absolute error, ($Sen$) Sensitivity, ($S_p$) Specificity, ($P_c$) Precision, ($F1$) F1 Score and ($Acc$) Accuracy. Each study used various techniques for a segmentation task. The final column shows the number of parameters for each model. Details of the error calculation are discussed in the text. A dash represents no available data and the best scores are shown in bold.

| Method | Sm | Ea | MAE | Sen | $S_p$ | $P_c$ | F1 | Acc | Parameters |
|---|---|---|---|---|---|---|---|---|---|
| ED-CNN (This work) | 0.82 ± 0.01 | 0.86 ± 0.01 | **0.0075 ± 0.0005** | 0.82 ± 0.05 | **0.996 ± 0.001** | 0.76 ± 0.04 | 0.77 ± 0.02 | **0.994 ± 0.002** | **49 K** |
| Two stream input encoder-decoder (*Elharrouss, Subramanian & Al-Maadeed, 2021*)) | — | — | 0.062 | 0.711 | 0.993 | 0.856 | 0.784 | — | — |
| Semi-inf-net (*Fan et al., 2020*) | 0.800 | 0.894 | 0.064 | 0.725 | 0.960 | — | — | — | 33,122 K |
| ADID-UNET (*Raj et al., 2021*) | **0.8509** | **0.9449** | 0.0082 | 0.7973 | 0.9966 | 0.8476 | 0.8200 | 0.9701 | 52,162 K |
| U-NET (*Raj et al., 2021*) | 0.8400 | 0.9390 | 0.0088 | 0.8052 | 0.9957 | 0.8247 | 0.8154 | 0.9696 | 56,223 K |
| RDAU-NET (*Zhuang et al., 2019*) | — | — | — | 0.8319 | 0.9934 | 0.8858 | 0.8478 | 0.9791 | — |
| RDAU-NET-WGAN (*Negi et al., 2020*) | — | — | — | **0.8837** | 0.9926 | **0.9117** | **0.8975** | 0.9808 | 28,817 K |

## Robustness of the prediction and training times

We investigated how reproducible the scores obtained for our model are, this is not often reported in the literature. To do so, the model was re-trained 10 separate times with a newly made training dataset. To achieve this, we take the original data and re-randomise the 80/20 split for the test and training data before training, as previously explained and shown in 3. The re-randomisation meant that the composition of each testing and training set was different. The new 80% data sample is then augmented to make the 5,900 images in the training dataset as shown in Figs. 3 and S2. Each time the model is run, its memory is cleared to ensure it does not retain information from the previous run, ensuring that the test set is never seen before being tested. Metric scores are computed (see evaluation metrics section) for each run and derive an average metric value with its corresponding error (All computed values are shown in Table S1).

To calculate the error with a confidence interval of 99%, we use small number statistics from the work of *Dean & Dixon (1951)*. First the range of each run is calculated and then multiply this by the recommended coefficient (Table 1, column 8, *n* = 10 of *Dean & Dixon (1951)*). This gives us the range multiplied by 0.33-thus an estimate of the variance of our results shown in Table 1.

We ran our dataset with the ADID-UNET model of *Raj et al. (2021)*, obtained from their supplementary information section to estimate comparative run times. When doing so we saw that their model took an average of 65.18 *s* per epoch with our dataset. Our model achieved a training time of only 19.5 *s* per epoch on the same dataset. When we look at the difference in parameters, the ADID-UNET model having 52,162 K and ours only having 49K, it explains why our model would be three times faster on average than theirs even when run on the same GPU and with the same dataset. This shows that our ED-CNN is indeed faster than other methods, yet yields similar results.

## DISCUSSION

Our ED-CNN model offers a simple model which successfully predicts contours/masks from CT scans indicating COVID-19 infection. Although our method does not always improve on the metrics reported in the literature using other techniques, it gives similar scores with much lower model complexity. Our model contains a total of 49,761 parameters, comparing this to other models that give similar results, and have total parameters exceeding 56 million. This difference of several orders of magnitude in the number of parameters between our model and leading methods, suggests a much reduced computational training cost.

As is the case for most deep-learning approaches, using a relatively small dataset means that the impact of low-quality images is magnified and could potentially bias some of the training. Therefore we explored the variability of our model training and the error analysis performed gives a more realistic picture of the capabilities of our approach.

There is success in that our model can be trained with a starting dataset as small as 1,844 images, this may be of great advantage in clinical studies where patient numbers may be low or when there may be a lack of readily available data. Our simple model offers the potential to be a fast and flexible tool that could be deployed efficiently in clinical settings where computational resources may be limited. Moreover, we anticipate further extension of our model to contouring different pathologies of the lung. For example, using CT scans from a patient with lung cancer, provided labelled datasets by oncologists are available, we can train to detect cancerous lesions instead of COVID-19 with simple retraining. The benefits of this approach include automatic screening of CT images to generate routine lesion predictions (tumour or mass). A fertile area of research and clinical development in radiotherapy concerns the daily adaption of treatment based on verification CT image acquired on the treatment machine, otherwise known as adaptive radiation therapy (ART), which can make great use of deep learning models to rapidly contour patients during treatment without them leaving the machine (*Glide-Hurst et al., 2021*). Our methodology could ultimately help speed up treatment in related areas, by rapidly evaluating how much a tumour has shrunk during treatment, or how much it has grown/spread if treatment is not working. This allows for a faster change in treatment plan which otherwise requires a multidisciplinary staff group to re-scan and re-contour the patient's lungs to determine what, if any, changes there were to their tumour. This is not commensurate with the requirements for daily adaptation workflow. Our approach could also potentially detect areas that may otherwise be missed when looking over the initial CT scan with this type of early-detection measures ultimately leading to a greater chance of survival.

## CONCLUSIONS

We have proposed a method for computer prediction of contours for CT scans for COVID-19 lung data that successfully predicts/segments infected areas. In comparison to other published techniques, our model is simpler whilst still performing on par with the literature. Our proposed model (ED-CNN) achieves similarity measures for $S_m = 0.82 \pm 0.01$, $E_a = 0.86 \pm 0.01$, $MAE = 0.0075 \pm 0.0005$, $Sen = 0.82 \pm 0.05$, $S_p = 0.996 \pm 0.001$, $P_c = 0.76 \pm 0.04$, $F1 = 0.77 \pm 0.02$, $Acc = 0.994 \pm 0.002$.

**Table A1  Key of symbols used.**

| Symbol | Definition |
| --- | --- |
| $\alpha$ | A constant set to 0.5 |
| $S_o$ | The target perception similarity |
| $S_p$ | The final prediction result |
| $G$ | The ground truth mask |
| $S_r$ | The regional perceptual similarity |
| $w$ | Width of $G$ |
| $h$ | Height of $G$ |
| $(x, y)$ | The coordinates of each pixel in $G$ |
| $\phi$ | The enhanced alignment matrix |

These scores show that our model is able to produce masks/contours from unseen CT images with a high level of similarity to the ground truth contours.

Two points are worth noting: firstly, our model achieves the best overall scores for $MAE = 0.0075 \pm 0.0005$, $S_p = 0.996 \pm 0.001$ and $Acc = 0.994 \pm 0.002$, suggesting that a simpler model can outperform existing approaches for some metrics. This opens up new avenues in exploring how computationally simple models could be a suitable alternative to standard deep models, such as U-nets. Secondly, we observe that there is very little variability for all metrics used. Indeed, the largest absolute variation (0.05) is observed for *Sen*, which corresponds to a 6% variation, while all other metrics have lower variations. This indicates that the model training workflow is robust and leads only to small variations in the prediction outcome—a desirable feature for possible usage in a clinical environment. Our results show that it is possible to achieve similar results to those of more complicated methods with a much smaller, less complex, model if care is taken to ensure it is appropriately trained as can be seen with our smaller number of parameters of 49,000 compared to others of 30–56 million.

Finally, our approach towards a lean yet efficient segmentation model opens the possibility of performing re-training using moderately-powered platforms typically available in a medical setting. We have shown that even a small dataset can lead to segmentation results comparable to those of more elaborate models, thus potentially allowing a more "personalised" medicine approach for various lung lesions (COVID-19 or cancer, for example) by using data from a smaller group of patients. Future work with this model will include looking at the robustness of the model, if it can withstand images of lower quality, what its limitations will be. We also plan to investigate how well the ED-CNN performs when applying it to patients with lung cancer once re-trained on a new dataset with a different lung lesion to what was used in this work.

## APPENDIX

The abbreviations used in this article are detailed in Table A1.

# ACKNOWLEDGEMENTS

We acknowledge the Viper High-Performance Computing facility of the University of Hull and its support team. We also acknowledge constructive comments by reviewers of an earlier draft of the article.

## Funding

Kiri S. Newson received funding from the Bell Burnell Scholarship fund administered by the Institute of Physics. Kiri S. Newson also received funding from The University of Hull. The funders had no role in study design, data collection and analysis, decision to publish, or preparation of the manuscript.

## Grant Disclosures

The following grant information was disclosed by the authors:
Institute of Physics.
The University of Hull.

## Competing Interests

The authors declare that they have no competing interests.

## Author Contributions

- Kiri S. Newson conceived and designed the experiments, performed the experiments, analyzed the data, performed the computation work, prepared figures and/or tables, authored or reviewed drafts of the article, and approved the final draft.
- David M. Benoit conceived and designed the experiments, authored or reviewed drafts of the article, and approved the final draft.
- Andrew W. Beavis conceived and designed the experiments, authored or reviewed drafts of the article, and approved the final draft.

## Data Availability

A copy of the code for ED-CNN is available at GitHub and Zenodo:

- https://github.com/Kiri-Newson/ED-CNN/blob/main/GitHub_version_ED_CNN_Model.ipynb.

- Kiri Newson. (2023). Kiri-Newson/ED-CNN: ED-CNN (ED-CNN). Zenodo. https://doi.org/10.5281/zenodo.8099854.

The data are also available at Zenodo:

Ma Jun, Ge Cheng, Wang Yixin, An Xingle, Gao Jiantao, Yu Ziqi, Zhang Minqing, Liu Xin, Deng Xueyuan, Cao Shucheng, Wei Hao, Mei Sen, Yang Xiaoyu, Nie Ziwei, Li Chen, Tian Lu, Zhu Yuntao, Zhu Qiongjie, Dong Guoqiang, & He Jian. (2020). COVID-19 CT Lung and Infection Segmentation Dataset (Verson 1.0) [Data set]. Zenodo. https://doi.org/10.5281/zenodo.3757476.

## Supplemental Information

Supplemental information for this article can be found online at http://dx.doi.org/10.7717/peerj-cs.2178#supplemental-information.

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
