# Peer review of "Encoder-decoder convolutional neural network for simple CT segmentation of COVID-19 infected lungs"

_PeerJ Computer Science, doi:10.7717/peerj-cs.2178_

## Round 0.1 · original submission · Minor Revisions

Dear authors,

Following your appeal, and after a meticulous study of the entire process regarding the Encoder-decoder convolutional neural network for simple CT segmentation publication proposal, I changed the status from Rejected to Review (major).

However, after checking some of the answers provided, I suggest that these can be reflected in the next version to be sent. I will list those that should be reconsidered/improved in relation to the answers you provided, without prejudice to your opinion.

Review 1
Take care of everything just like you stated in the response letter.
* * *
Review 2
Experimental Design:
2) “Please provide a flowchart of the methodology. ‎This will help the reader to get a better understanding of what is going on in the proposed ‎system.‎“
I think you missed the reviewer point. The request is an overall diagram of the entire system, including the model (fig 2), the information of data augmentation as well. Google for pipelines applied to these subjects to get some ideas… with data collection, data preparation, model selection, etc.

By the way, flowcharts have rules, my advice is for you to search for flowchart symbols and notation. Fig. 3 is not a flowchart.
* * *
Review 3
These are the ones that you should address more carefully. Take care of everything just like you stated in the response letter, but I’d highlight the following:

Experimental design
15) and 16)
Provide further details of how to configure the parameters until you get the “best” setup/model. Prove your claims with charts or whatever supports your decisions.
12)
Same as mentioned above for reviewer 2.

Validity of the findings
22)
Same as mentioned above for reviewer 2, and 12) for reviewer 3.
20) “The manuscript does not provide a clear contribution to the field of research.”
Address this in Introduction, it is important.
25) “The conclusions in this manuscript are primitive. Please, write your conclusions.”
Elaborate scientifically the conclusions.
26) “There are no citations for many sentences in the manuscript. Why? Please check.”
The same that you answered in the reviewer 2, in the basic reporting, 2). Your claims with references that support them.
* * *
Additionally, I detected some minor errors that can be easily corrected:
Eq 6 -> Sen instead of MAE
Table 2 results are in seconds, right?

· Appeal

Appeal

We strongly disagree with the decision to reject our manuscript “Encoder-Decoder Convolutional Neural Network for Simple CT Segmentation” (CS-2023:07:88689:0:2:X). We previously addressed all the comments in the first round of reviews and we feel that the latest review is limited to style recommendations and does not indicate any major flaws in contents of the paper. We attach a point by point response to the new reviewers comments and would strongly urge you to reconsider the decision.


· · Academic Editor

Reject

Reviewer comments are against the publication of the manuscript. Therefore I must reject it.

Reviewer 1 ·

Basic reporting

no comment

Experimental design

no comment

Validity of the findings

no comment

Additional comments

no comment

Annotated reviews are not available for download in order to protect the identity of reviewers who chose to remain anonymous.

Reviewer 2 ·

Basic reporting

In this paper, authors present the application of an Encoder-Decoder convolutional neural network (ED-CNN) model to segment COVID-19 computerised tomography (CT) data. They show that their simple approach achieves results comparable to those of previously published studies, which use more complex deep-learning networks.
Authors did an interesting work and we appreciated their efforts in manuscript revision as well. A few following concerns need to be discussed and revised carefully before the paper's acceptance.

1 The abstract section is inconsistent and does not reflect the main contributions of the manuscript. The authors should rewrite the abstract section to mention primary contributions, experimental results, and global implications.

2 When writing phrases like “used stacked sparse auto-encoders to classify prostate cancer”, it must discuss recent related works in order to sustain the statement that authors miss: 10.1155/2023/2345835; 10.1155/2022/2665283

3 What is justification of “Simple CT Segmentation” in Title?

4 The main contributions of the manuscript are not clear. The main contributions of the ‎article must be very clear and would be better if summarize ‎them into 3-4 points at the ‎end of the introduction.‎


5 An introduction is an important road map for the rest of the paper and should consist of an opening hook to catch the researcher's attention, relevant background studies, and a concrete statement that presents the main argument, but your introduction lacks these fundamentals, especially relevant background studies. This related work is just listed out without comparing the relationship between this paper's model and theirs; only the method flow is introduced at the end, and the principle of the method is not explained. To make soundness of your study must include these latest related works and discuss them.
I (2023). Few-Shot Class-Incremental Learning for Medical Time Series Classification. IEEE Journal of Biomedical and Health Informatics. doi: 10.1109/JBHI.2023.3247861
II (2023). Exploring the Computational Effects of Advanced Deep Neural Networks on Logical and Activity Learning for Enhanced Thinking Skills. Systems, 11(7), 319. doi: 10.3390/systems11070319"
III (2022). An Effective WSSENet-Based Similarity Retrieval Method of Large Lung CT Image Databases. KSII Transactions on Internet & Information Systems, 16(7). doi: 10.3837/tiis.2022.07.013
IV (2023). Iterative reconstruction of low-dose CT based on differential sparse. Biomedical Signal Processing and Control, 79, 104204. doi: 10.1016/j.bspc.2022.104204
V (2023). Soft Tissue Feature Tracking Based on Deep Matching Network. Computer Modeling in Engineering & Sciences, 136(1), 363-379. doi: 10.32604/cmes.2023.025217

Experimental design

1 “We follow a supervised training protocol”, what is logic behind this strategy?

2 Please provide a flowchart of the methodology. ‎This will help the reader to get a better understanding of what is going on in the proposed ‎system.‎

3 How to optimize these hyperparameters during model training?

Validity of the findings

1 Authors should mention implementation challenges.

2 Moreover, it should be noticed that the clinical appliance has to be decided by medicals since the existing differences between the real image and the one generated by the proposed model could be substantial in the medical field.

·

Basic reporting

The manuscript presented “a CT segmentation approach using encoder-decoder CNN”. However, the major and critical weak points are that:
(1) Their proposed work discussion is weak distributed to be described or analyzed.
(2) The novelty is not guaranteed.
(3) Their work is not compared with state-of-the-art approaches nor related studies.
(4) Their experiments leak from the descriptive and statistical analysis.

Experimental design

(12) [METHODOLOGY] Where is the overall pseudocode? Flowchart? of the suggested approach?
(13) [EXPERIMENTS] The working environment (i.e., software and hardware) should be declared and added to a table.
(14) [EXPERIMENTS] The experimental configurations (i.e., settings) should be declared and added to a table.
(15) [EXPERIMENTS] What are the criteria for selecting the experimental configurations?
(16) [EXPERIMENTS] More experiments should be conducted using different configurations.
(17) [EXPERIMENTS] Why did not the authors compare their approach with others in a table?
(18) [EXPERIMENTS] Why did not the authors compare their approach with another approach to compare the suggested approach efficiency and applicability?
(19) [EXPERIMENTS] Where is the detailed and statistical discussion of the reported results?

Validity of the findings

(10) [METHODOLOGY] The suggested approach is not clearly discussed. More scientific details should be added.
(11) [METHODOLOGY] What are the used equations in the suggested approach? In other words, how the suggested approach is derived?
(12) [METHODOLOGY] Where is the overall pseudocode? Flowchart? of the suggested approach?
(20) The manuscript does not provide a clear contribution to the field of research.
(21) The manuscript does not provide a clear justification for the research.
(22) [RESULTS] The authors have not provided adequate visual aids, such as graphs or tables, to help readers understand the data.

Additional comments

Journal: PeerJ
Manuscript Title: Encoder-decoder convolutional neural network for simple CT segmentation
Manuscript ID: PeerJ 88689v1
Submission Date: Monday, September 11, 2023
The manuscript presented “a CT segmentation approach using encoder-decoder CNN”. However, the major and critical weak points are that:
(1) Their proposed work discussion is weak distributed to be described or analyzed.
(2) The novelty is not guaranteed.
(3) Their work is not compared with state-of-the-art approaches nor related studies.
(4) Their experiments leak from the descriptive and statistical analysis.
The rest of my review presents other weak points, comments, and opinions in detail.
Overall Comments:
(1) [ABSTRACT] The abstract should contain the best-achieved results from the performed experiments.
(2) [ABSTRACT] The abstract should reflect the contributions of the manuscript. I suggest rewriting it.
(3) [INTRODUCTION] The authors should provide a clear problem definition and contributions in the introduction section.
(4) [RESEARCH QUESTION] Where is the research question and research gap?
(5) [RESEARCH QUESTION] The research question is not well-formulated or is poorly motivated, and the paper does not provide new insights or information that is not already known.
(6) [RELATED WORK] Where are the related studies? They should be declared in a separate section.
(7) [RELATED WORK] A table of comparisons should be added at the end of the related studies section to praise the pros. and cons. of them. The year column should be added and they should be ordered by it.
(8) [EQUATIONS] The authors should follow the journal authors’ guidance in writing the equations, symbols, and variables. Please, refer to the authors guidelines on the journal official website.
(9) [EQUATIONS] Where are the equations of the used metrics?
(10) [METHODOLOGY] The suggested approach is not clearly discussed. More scientific details should be added.
(11) [METHODOLOGY] What are the used equations in the suggested approach? In other words, how the suggested approach is derived?
(12) [METHODOLOGY] Where is the overall pseudocode? Flowchart? of the suggested approach?
(13) [EXPERIMENTS] The working environment (i.e., software and hardware) should be declared and added to a table.
(14) [EXPERIMENTS] The experimental configurations (i.e., settings) should be declared and added to a table.
(15) [EXPERIMENTS] What are the criteria for selecting the experimental configurations?
(16) [EXPERIMENTS] More experiments should be conducted using different configurations.
(17) [EXPERIMENTS] Why did not the authors compare their approach with others in a table?
(18) [EXPERIMENTS] Why did not the authors compare their approach with another approach to compare the suggested approach efficiency and applicability?
(19) [EXPERIMENTS] Where is the detailed and statistical discussion of the reported results?
(20) The manuscript does not provide a clear contribution to the field of research.
(21) The manuscript does not provide a clear justification for the research.
(22) [RESULTS] The authors have not provided adequate visual aids, such as graphs or tables, to help readers understand the data.
(23) [ABBREVIATIONS] The authors should add a table of abbreviations in the revised manuscript.
(24) [SYMBOLS] The authors should add a table of symbols in the revised manuscript.
(25) [CONCLUSIONS] The conclusions in this manuscript are primitive. Please, write your conclusions.
(26) [REFERENCES] There are no citations for many sentences in the manuscript. Why? Please check.
(27) [REFERENCES] The references should be written in the same style following the journal authors’ guidance.
(28) [REFERENCES] Recent citations from 2021 to 2023 should be added to the manuscript.
(29) [PROOFING] The authors should get editing help from someone with full professional proficiency in English.
(30) [PROOFING] The manuscript should be checked again to fix any typos such as missing spaces and commas.
(31) [CONSISTENCY] The manuscript structure is too short. It must be elaborated in their applied technology as should support more rigorous technical aspects.
(32) [CONSISTENCY] Some paragraphs are wrapped in more than 10 lines. They should be split concisely.
(33) [NOVELTY] What is the novelty of the suggested approach?
(34) [LIMITATIONS] What are the limitations of the current study? It should be added in a separate section.
For the authors in case of the authors got a chance to review the manuscript and submit the revised one after the editor’s decision, please, provide a table in the revised manuscript mentioning (1) the comment, (2) the authors’ response, and (3) the authors’ change (if applicable). Please, consider all of the comments and don’t ignore any of them.
Please, refer to the attached file "PeerJ 88689v1 Reviewer.pdf" for the same comments in an organized format.

---

## Round 0.2 · Major Revisions

Dear authors,

You are advised to critically respond to all comments point by point when preparing a new version of the manuscript and while preparing for the rebuttal letter. Please address all the comments/suggestions provided by the reviewers.

Kind regards,
PCoelho

Reviewer 1 ·

Basic reporting

The authors addressed the issues I highlighted, I have no further comments.

Experimental design

The authors addressed the issues I highlighted, I have no further comments.

Validity of the findings

The authors addressed the issues I highlighted, I have no further comments.

Reviewer 2 ·

Basic reporting

no comment

Experimental design

no comment

Validity of the findings

no comment

Additional comments

Authors have answered all my questions satisfactorily.

·

Basic reporting

Journal: PeerJ Computer Science
Manuscript Title: Encoder-decoder convolutional neural network for simple CT segmentation
Manuscript ID: 88689v2
Submission Date: Monday, January 22, 2024
The manuscript presented “a CT segmentation approach using encoder-decoder CNN”. However, the major and critical weak points are that:
(1) Their proposed work discussion is weak distributed to be described or analyzed.
(2) The novelty is not guaranteed.
(3) Their experiments leak from the descriptive and statistical analysis.

Experimental design

(1) [ABSTRACT] The abstract should reflect the contributions of the manuscript. I suggest rewriting it.
(2) [INTRODUCTION] The authors should provide a clear problem definition and contributions in the introduction section.
(3) [RESEARCH QUESTION] Where is the research question and research gap?
(4) [RESEARCH QUESTION] The research question is not well-formulated or is poorly motivated, and the paper does not provide new insights or information that is not already known.
(5) [RELATED WORK] A table of comparisons should be added at the end of the related studies section to praise the pros. and cons. of them. The year column should be added and they should be ordered by it.
(6) [EQUATIONS] The authors should follow the journal authors’ guidance in writing the equations, symbols, and variables. Please, refer to the authors guidelines on the journal official website.
(7) [EQUATIONS] Where are the equations of the used metrics?
(8) [METHODOLOGY] The suggested approach is not clearly discussed. More scientific details should be added.
(9) [METHODOLOGY] What are the used equations in the suggested approach? In other words, how the suggested approach is derived?
(10) [METHODOLOGY] Where is the overall pseudocode? Flowchart? of the suggested approach?

Validity of the findings

(11) [EXPERIMENTS] The working environment (i.e., software and hardware) should be declared and added to a table.
(12) [EXPERIMENTS] The experimental configurations (i.e., settings) should be declared and added to a table.
(13) [EXPERIMENTS] What are the criteria for selecting the experimental configurations?
(14) [EXPERIMENTS] More experiments should be conducted using different configurations.
(15) [EXPERIMENTS] Why did not the authors compare their approach with others in a table?
(16) [EXPERIMENTS] Why did not the authors compare their approach with another approach to compare the suggested approach efficiency and applicability?
(17) [EXPERIMENTS] Where is the detailed and statistical discussion of the reported results?
(18) The manuscript does not provide a clear contribution to the field of research.
(19) The manuscript does not provide a clear justification for the research.
(20) [RESULTS] The authors have not provided adequate visual aids, such as graphs or tables, to help readers understand the data.

Additional comments

(21) [ABBREVIATIONS] The authors should add a table of abbreviations in the revised manuscript.
(22) [CONCLUSIONS] The conclusions in this manuscript are primitive. Please, write your conclusions.
(23) [REFERENCES] There are no citations for many sentences in the manuscript. Why? Please check.
(24) [REFERENCES] The references should be written in the same style following the journal authors’ guidance.
(25) [REFERENCES] Recent citations from 2021 to 2023 should be added to the manuscript.
(26) [PROOFING] The authors should get editing help from someone with full professional proficiency in English.
(27) [PROOFING] The manuscript should be checked again to fix any typos such as missing spaces and commas.
(28) [CONSISTENCY] The manuscript structure is too short. It must be elaborated in their applied technology as should support more rigorous technical aspects.
(29) [CONSISTENCY] Some paragraphs are wrapped in more than 10 lines. They should be split concisely.
(30) [NOVELTY] What is the novelty of the suggested approach?
(31) [LIMITATIONS] What are the limitations of the current study? It should be added in a separate section.

---

## Round 0.3 · Minor Revisions

Dear authors, we are pleased to verify that you meet most of the reviewer's valuable feedback to improve your research. Nevertheless, some minor issues are still pending, so please address them accordingly.

Reviewer 2 ·

Basic reporting

Clear

Experimental design

Good

Validity of the findings

Conclusions are well stated.

·

Basic reporting

Journal: PeerJ Computer Science
Manuscript Title: Encoder-decoder convolutional neural network for simple CT segmentation
Manuscript ID: PeerJ 88689v3
Reviewer Number: 3
Submission Date: Wednesday, April 3, 2024
The authors have made a revision. However, there are some concerns that should be addressed.

Experimental design

Overall Comments:
(1) The manuscript should be checked again to fix any typos. For example, [‘person-to-person transmission’] should be double quotes to match the rest of the paper. Unify the single or double quotes.
(2) Figure 1: In [CT images/contours used within figure by Jun et al. (2020)] where are the contours?
(3) In the paragraph starting with “Several studies have explored …”, the mentioned studies should be written in the past tense. For example [Zhuang et al. (2019) proposes a Residual-Dilated-Attention-Gate-U-Net (RDAU-NET) model]. Also, this paragraph is two long, split it in two concise paragraphs.
(4) What are the hypotheses behind this study?
(5) Figure 1 and its paragraph should be moved to a “Materials” subsection in the Methods section.
(6) Please, avoid the usage of “We”, “Our” etc. in the manuscript.
(7) In [Abbreviations used throughout are summarised in Table A1.] Are they abbreviations or symbols?
(8) Describing the evaluation metrics such as accuracy and precision can be omitted. Equations can be summarized in a table.
(9) The authors should discuss the complexity analysis of their approach.
(10) What is the superiority of the current study compared to the related studies?

Validity of the findings

.

Additional comments

For the authors in case of the authors got a chance to review the manuscript and submit the revised one after the editor’s decision, please, provide a table in the revised manuscript mentioning (1) the comment, (2) the authors’ response, and (3) the authors’ change (if applicable). Please, consider all of the comments and don’t ignore any of them.
Please, refer to the attached file "PeerJ 88689v3 Reviewer.pdf" for the same comments in an organized format.

---

## Round 0.4 · Minor Revisions

Dear authors,

Thanks a lot for your efforts to improve the manuscript, although some important issues are still requiring your attention, as very well stated by the reviewer #3.

Kind regards,
PCoelho

Reviewer 2 ·

Basic reporting

no comment

Experimental design

no comment

Validity of the findings

no comment

·

Basic reporting

The authors propose a CT segmentation approach that requires applying their method to various datasets and comparing the results with related studies. Although this intention is stated in the abstract, the title is misleading. The authors should update the title to include "COVID-19". Additionally, they should apply their approach to different COVID-19 CT datasets. There are many public datasets available for research.

Furthermore, in the manuscript, the authors state: "We chose to split this into 20% (369 images) as test data, which was set aside for later evaluation of the model. It is crucial to split the data before any augmentation or manipulation to make sure it is a true test of the model and ensure the model has not seen another version of the image in the training process." While I agree with the data augmentation aspect, there appears to be data leakage in the preprocessing step. In Figure 3, the dataset consists of 3,520 images from 20 cases (see Materials Section). During the train/test split, data leakage occurs across different cases.

Experimental design

.

Validity of the findings

.

---

## Round 0.5 · accepted · Accept

Dear authors, we are pleased to verify that you meet the reviewer's valuable feedback to improve your research.

Thank you for considering PeerJ Computer Science and submitting your work.

·

Basic reporting

The authors have addressed many of the suggested comments, and considering the effort put into the manuscript, I recommend its acceptance.

Experimental design

.

Validity of the findings

.

Additional comments

The authors may need to proofread the manuscript to correct any typos or language errors.